# Concomitant Hepatic Artery Resection for Advanced Perihilar Cholangiocarcinoma: A Narrative Review

**DOI:** 10.3390/cancers14112672

**Published:** 2022-05-27

**Authors:** Takehiro Noji, Satoshi Hirano, Kimitaka Tanaka, Aya Matsui, Yoshitsugu Nakanishi, Toshimichi Asano, Toru Nakamura, Takahiro Tsuchikawa

**Affiliations:** Department of Gastroenterological Surgery II, Faculty of Medicine, Hokkaido University, Sapporo City 060-8638, Japan; satto@med.hokudai.ac.jp (S.H.); kimitaka.t@gmail.com (K.T.); a_ma.sur@hotmail.co.jp (A.M.); y.nakanishi@mac.com (Y.N.); toasa0616@gmail.com (T.A.); torunakamura@med.hokudai.ac.jp (T.N.); tsuchi-t@med.hokudai.ac.jp (T.T.)

**Keywords:** perihilar cholangiocarcinoma, concomitant hepatic artery resection, prognosis

## Abstract

**Simple Summary:**

In the eighth edition of their cancer classification system, the Union for International Cancer Control (UICC) defines contralateral hepatic artery invasion as T4, which is considered unresectable as it is a “locally advanced” tumour. However, in the last decade, several reports on hepatic artery resection (HAR) for perihilar cholangiocarcinoma (PHCC) have been published. The reported five-year survival rate after HAR is 16–38.5%. Alternative procedures for the treatment of HAR have also been reported. In this paper, we review HAR for PHCC, focusing on its history, diagnosis, procedures, and alternative procedures.

**Abstract:**

Perihilar cholangiocarcinoma (PHCC) is one of the most intractable gastrointestinal malignancies. These tumours lie in the core section of the biliary tract. Patients who undergo curative surgery have a 40–50-month median survival time, and a five-year overall survival rate of 35–45%. Therefore, curative intent surgery can lead to long-term survival. PHCC sometimes invades the surrounding tissues, such as the portal vein, hepatic artery, perineural tissues around the hepatic artery, and hepatic parenchyma. Contralateral hepatic artery invasion is classed as T4, which is considered unresectable due to its “locally advanced” nature. Recently, several reports have been published on concomitant hepatic artery resection (HAR) for PHCC. The morbidity and mortality rates in these reports were similar to those non-HAR cases. The five-year survival rate after HAR was 16–38.5%. Alternative procedures for arterial portal shunting and non-vascular reconstruction (HAR) have also been reported. In this paper, we review HAR for PHCC, focusing on its history, diagnosis, procedures, and alternatives. HAR, undertaken by established biliary surgeons in selected patients with PHCC, can be feasible.

## 1. Introduction

Perihilar cholangiocarcinoma (PHCC), also known as a Klatskin tumour, is one of the most intractable gastrointestinal malignancies [1,2]. These tumours are located in the core section of the biliary tract. They can cause obstructive jaundice and liver failure following cholangitis. Recent developments in antitumor therapy (chemotherapy) can improve the condition of patients with unresectable biliary malignancy. However, the reported median survival time (MST) is only 11–13 months [1]. Conversely, patients undergoing curative surgery have a 40–50-month MST and a five-year overall survival rate in 35–45% of patients. Therefore, curative intent surgery can lead to long-term survival [1,2].

PHCC sometimes invades the surrounding tissues, such as the portal vein, hepatic artery, perineural tissues around the hepatic artery, and hepatic parenchyma [3]. As surgery for PHCC requires multiorgan resection, such as major hepatectomy, extrahepatic bile duct resection, and lymphadenectomy of the hepatoduodenal ligament, it is considered to be one of the most demanding procedures and carries a high risk [4,5,6]. Therefore, morbidity and mortality rates are high in established high-volume centres. Recent reports on PHCC from these specialised centres revealed that postoperative morbidity was over 50% and mortality was 0–17% [7].

Before 2000, when the use of portal vein embolisation (PVE) became widespread and a major hepatectomy could be performed safely, postoperative liver failure was a serious problem [4]. Several “reduced surgeries” have been developed, such as resection of the hilar and periportal liver, including the caudate lobe and extended extrahepatic bile duct resection (hilar plate resection) [8]. In 2004, Kondo et al. [9] showed that a major hepatectomy with extrahepatic bile duct resection(s) (Hx with EBDR) was superior to reduced surgery in patients with advanced PHCC. Ikeyama et al. [10] also clearly showed that Hx with EBDR was a superior procedure compared with reduced surgery in patients with Bismuth type I or II PHCC. Currently, many established biliary surgeons advocate Hx with EBDR, and regional lymphadenectomy is the standard procedure for curative resection of PHCC, as it leads to en bloc resection of the intrahepatic bile ducts and hepatic portal plate tissue [1]. However, refinements in perioperative management and surgical technique in recent decades could not improve long-term survival. In addition to pathological factors, such as nodal metastasis or vascular invasion, cancer-free resection (R0) is the most important factor for long-term survival in patients with PHCC [6]. To achieve R0 resection, several extended surgical procedures, such as hepatopancreatoduodenectomy, concomitant portal vein resection (PVR), and trisectionectomy are now routinely performed worldwide [6,11,12,13]. Moreover, the Union for International Cancer Control’s (UICC) cancer classification, eighth edition, defines contralateral hepatic artery invasion as T4. Therefore, many biliary surgeons consider advanced PHCC cases requiring hepatic artery resection (HAR) to be unresectable due to the “locally advanced” nature of the tumour [14]. However, in the last decade, several reports on HAR for PHCC have been published, mainly by Japanese and Chinese surgeons [3,5,12,15,16,17,18,19,20,21]. Alternative procedures for HAR have also been reported [22].

In this article, we review HAR for PHCC, focusing on its history, diagnosis, procedures, and alternatives.

## 2. Anatomical Features of the Hepatic Hilum and Hepatic Artery

The right hepatic artery is closely associated with the posterior or anterior surface of the biliary confluence, and is often involved in tumours [3]. Previous research on surgical anatomy showed that the right hepatic artery, which runs on the right or dorsal side of the bile duct, had been replaced in approximately 25% of cases [23], and usually had no tumour invasion. In other words, 75% of patients had hepatic inflow vessels (the hepatic artery and portal vein) close to each other in the hilar area and were in contact with the hilar bile duct (Figure 1).

The bile duct wall in the hepatic hilum is very thin. As a result, the tumour easily invades the area outside of the bile duct, and spreads along the hepatic arterial plexus and other structures. Therefore, even if it does not directly invade the arterial wall structures, it is clinically in close proximity and requires HAR. In this case, patients with advanced perihilar cholangiocarcinoma, who required both HAR and PVR, are involved. A preoperative computed tomography (CT) reveals a tumour close to the right hepatic artery (Figure 2).

## 3. History of Concomitant HAR for PHCC (from the First Reported Case to 2009)

Table 1 shows the reports of HAR for PHCC before 2009. The first cases were reported in 1983 by Tsuzuki et al. [24] (Keio University, Tokyo, Japan). They reported two cases of left hepatectomy with concomitant hepatic artery resection and portal vein resection. Surgical results were reported in the Japanese literature, instead of the English. These patients survived for 18 months and 15 months, respectively. To the best of our knowledge, secondary cases of HAR in the English literature were reported by Lee et al. [25] (four cases) in 2000. However, detailed clinical information regarding these HAR cases is lacking. In 2001, Yamanaka et al. [26] reported 10 cases of HAR with or without portal vein resection between 1980 and 1998. They reported that one of the 10 patients who underwent HAR died after surgery. Details of prognoses of patients who underwent HAR are not shown. In 2003, Shimada et al. [27] reported their surgical results for a consecutive series of PHCC and gallbladder carcinoma. They performed six HAR, and six HAR with portal vein resection. One patient died after HAR with PVR. Cumulative three- and five-year survival rates after HAR for PHCC (with or without PVR) were 32% and 18%, respectively. Moreover, survival rates in patients with gallbladder carcinoma were very poor (all patients died within 18 months of surgery). Miyazaki et al. [28] reported a series of PHCC cases in 2007. They performed HAR in nine cases; however, three of the nine patients met with hospital death, which raised doubts regarding the feasibility of HAR in PHCC surgery. During this period, Kondo et al. of our department also reported two cases of HAR without artery reconstruction (using arterial portal shunting [APS]) for PHCC. APS is reviewed in another section. Although HAR for PHCC was performed by several established biliary surgeons during this period, the results were unsatisfactory.

## 4. History and Current Status of HAR for PHCC (from 2010 to the Present)

Surgical results and outcomes improved after 2010. In 2010, Nagino et al. [15] reported the consequences of 50 concomitant PVR and HAR cases. Their results (morbidity: 50% and mortality: 2%) were similar to those of non-vascular resection for PHCC. The cumulative five-year survival rate was 30%, which was better than that of non-surgical therapies. Following this report, several authors have documented their experience with HAR. Most cases have been reported in the East (Japan or China) [3,5,12,15,16,17,18,19,20,21]. The reported data are listed in Table 2. The morbidity and mortality rates were almost equivalent to those of non-vascular resection cases (16.1–69.9%). Mortality was the same or higher than that in non-vascular cases (0–15.4%). Shindo et al. [21] did not record long-term survival; however, the others showed 16–38.5% five-year survival rates after HAR. These reports showed that HAR in selected cases could lead to long-term survival in patients with T4 disease (with contralateral hepatic artery involvement). 

## 5. Procedure for HAR and Vascular Reconstruction

Appendix A shows our procedure for HAR in PHCC. The first step of the surgical procedure is similar to the standard resection for PHCC: lymphadenectomy (skeletonisation) of the hepatoduodenal ligament and distal bile duct resection just above the posterior superior pancreatic duodenal artery. We usually pick up the right hepatic artery (or posterior branch) at Rouviere’s sulcus before skeletonisation around the proper hepatic artery to confirm resectability. Hepatic artery reconstruction is usually performed as the final step after bile duct and portal vein resection and reconstruction. Based on the most recent report on HAR, 100 of 146 (68.5%) patients required concomitant portal vein resection.

Figure 3a,b show schemas of the hepatic artery resection and reconstruction by Nagino et al. [15]. To reconstruct hepatic artery flow, several techniques can be selected: end-to-end anastomosis, side-to-end anastomosis, and graft interposition. Some patients required the artery rotation technique (gastroduodenal, left hepatic, left or right gastric, and splenic arteries). The Nagoya University group mainly used end-to-end anastomosis (knot suturing using the microscopic vascular anastomosis technique by plastic surgeons). The Ogaki Municipal Hospital group reported the continuous suture method performed by general surgeons [32]. The prophylactic administration of anticoagulants and/or vasodilating agents is not routinely used.

## 6. Alternative Procedures for Arterial Reconstruction

### 6.1. Arterial Portal Shunting (APS)

APS was first described in 1989 by Sheli et al. [33] in the field of liver transplantation (LT). They shunted the recipient’s iliac artery to the donor portal vein using a vascular catheter. Bonner et al. [34] advocated that APS be used as a rescue procedure for cases of pre-existing diffuse portal vein thrombosis, a notable contraindication to orthotopic LT, or portal vein thrombosis complications.

We have advocated APS as an alternative procedure for microvascular arterial anastomosis for HAR in hepatopancreatic biliary surgery [35]. Suzuki et al. [36] performed an experimental study using a canine model and demonstrated that APS improved biliary oxygen saturation after hepatobiliary dearterialisation. In 2000, Kodo from our department reported findings similar to our initial results in a patient who had undergone left hepatectomy where APS was undertaken. This occurred during an emergency operation to salvage a dearterialised hepatobiliary system, which resulted from an accidental injury to a right hepatic artery encased by the tumour [35]. Nagino et al. [15] reported the use of APS as an alternative procedure for microvascular anastomosis in 2 of 50 consecutive hepatic artery reconstruction cases. Chen et al. [37] reported four cases of PHCC with APS. They concluded that APS with restriction of the arterial calibre appears to be a feasible and safe alternative for microvascular reconstruction after hepatic artery resection during radical surgery for PHCC.

From 2000 to 2007, we reported the results of APS in 18 patients with PHCC [22]. We confirmed collateral arterial flow to the remnant liver by angiography and embolised the APS 1 month after surgery to prevent portal vein hypertension. We encountered seven cases of multiple refractory liver abscesses among the 18 patients. Our results suggest that APS causes insufficient biliary arterialisation, resulting in liver abscess formation. Moreover, one patient who failed shunt embolisation died of gastrointestinal bleeding due to portal hypertension. Previous clinical and experimental reports have shown that APS has the disadvantage of leaving portal hypertension unchanged. Several authors have suggested that over-arterialisation of the liver eventually leads to fibrosis, portal vein aneurysm, portal hypertension, and liver fibrosis after APS [38]. Therefore, APS should be indicated when an artery cannot be anastomosed microscopically.

Performing the procedure for salvage after postoperative complications has also been reported by a few authors [34,39]. Basso et al. [39] reported a right hepatic arterial pseudoaneurysm after left hepatectomy for PHCC. They treated the pseudoaneurysm with arterial embolisation and performed APS because there was no flow in the intrahepatic artery, and reconstruction of the hepatic artery was impossible. They created an APS using the ileocolic artery and vein. This would have been easier than the original procedure (end-to-side anastomosis using the common hepatic artery and portal vein).

### 6.2. Non-Vascular Reconstruction

Another alternative procedure after HAR is non-vascular reconstruction. Peng et al. [18] reported 26 cases of HAR. The hepatic artery was not reconstructed after the HAR. However, the number of cases without non-arterial reconstruction was unclear. They advocated that non-arterial reconstruction should be performed when (a) the tumour had infiltrated the hepatic artery with disappearance or markedly reduced arterial flow, as detected by intraoperative ultrasound; and (b) if the colour of the liver did not change by visual observation when the hepatic artery was blocked for 5 min. Hu et al. [19] reported the results of 63 HAR patients, in which HAR without hepatic artery reconstruction was performed in 34 patients. The case selection criteria for non-arterial reconstruction were similar to those of Peng et al. [18]. There were no significant differences with or without arterial reconstruction (Table 2).

It is well known that arterial collateral flow from arteries around the liver, such as the right inferior diaphragmatic artery and/or adrenal gland artery, can provide intrahepatic arterial flow when native hepatic arterial flow is absent. Therefore, HAR without arterial reconstruction is a possible option in selected cases.

## 7. Associating Liver Partition and Portal Vein Ligation for Staged Hepatectomy Procedures and Liver Transplantation for Unresectable PHCC

PHCC with insufficient remnant liver volume, and highly advanced vascular invasion not amenable to reconstruction, are clearly considered unresectable. However, associating liver partition and portal vein ligation for staged hepatectomy (ALPPS) is a potential option for effectively increasing remnant liver volume to allow resection. ALPPS achieves fast hypertrophy of the future liver remnant (74% after a median of 9 days), avoiding the risk of tumour progression compared with PVE [40]. However, ALPPS procedures for PHCC are reportedly associated with significant mortality. Olthof et al. reported a 48% (14/29) 90-day mortality rate for this approach compared to only 13% in 257 patients who underwent major liver resection without ALPPS. They concluded that ALPPS should not be recommended for PHCC.

The ideal option for unresectable PHCC would be liver transplantation (LT). A few early studies failed to show promising results after LT for PHCC, with three- and five-year survival rates never exceeding 40% and 30%, respectively, and with very high tumour recurrence rates [41,42,43]. In 2016, a retrospective study was undertaken by the European Liver Transplant Registry, which considered patients receiving transplants for h-CCA between 1990 and 2010. They compared 28 patients included in the “Mayo protocol” with 77 not included in the protocol. In the overall cohort of 105 patients, an actuarial five-year survival rate of 32% was observed. The patients within the Mayo Clinic (Rochester, MN, USA) criteria showed significantly better five-year survival rates than those who were not (59% vs. 21%; *p* = 0.001) [41]. The history and future of LT for PHCC have been described in a recent review [41].

## 8. Future Vision

One of the major causes of mortality after PHCC surgery is liver failure [7], against which left hepatectomy may provide superior protection compared to right hepatectomy. A recent study undertaken by our group showed that left hepatectomy for Bismuth type I/II PHCC achieved equivalent survival rates compared to right hepatectomy [44]. Sugiura et al. showed that left hepatectomy with concomitant arterial resection is a valid alternative to right hepatectomy for Bismuth type I/II PHCC in patients with insufficient left liver functional reserve, with no significant difference in survival [45]. These data suggested that HAR could be performed not only for advanced PHCC with hepatic arterial invasion, but also Bismuth type I/II PHCC, to help prevent post-hepatectomy liver failure. Chemotherapy in addition to LT would also be an alternative to HAR for PHCC.

## 9. Conclusions

Recent developments in operative procedures and perioperative management may have made radical resection of T4 PHCC with contralateral hepatic artery involvement more amenable. Hepatic resection with contralateral HAR for locally advanced PHCC is feasible in selected patients at high-volume centers.

## Figures and Tables

**Figure 1 cancers-14-02672-f001:**
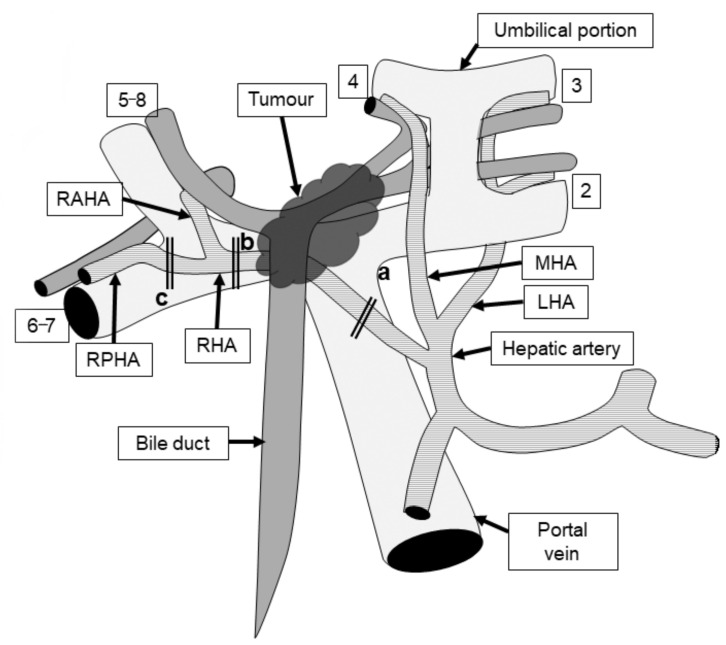
A schema of the anatomical relationship between the bile duct, hepatic artery, portal vein, and a Bismuth type 3B tumour in text-book-type anatomy. In left hepatectomy with hepatic artery resection (HAR), the right hepatic artery is cut proximal (**a**) and distal (**b**) to the tumour. In left trisectionectomy, the distal cut end of the hepatic artery can be placed further from the tumour at the right posterior hepatic artery (**c**). RHA: right hepatic artery; MHA: middle hepatic artery; LHA: left hepatic artery; RAHA: right anterior hepatic artery; and RPHA: right posterior hepatic artery. Numerals represent Cournand’s hepatic segments which the hepatic inflow vascular structures perfuse. (Reproduced with permission from Ref. [3]. 2022, LWW).

**Figure 2 cancers-14-02672-f002:**
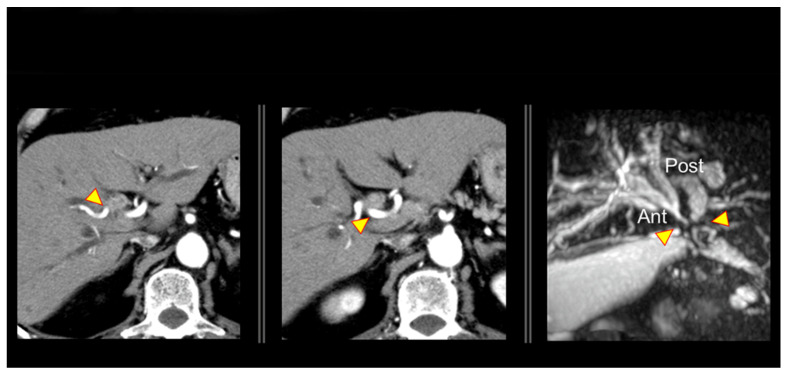
Advanced case of perihilar cholangiocarcinoma (T4N1M0 Stage IIIc) that required left hepatectomy with concomitant hepatic artery and portal vein resection.

**Figure 3 cancers-14-02672-f003:**
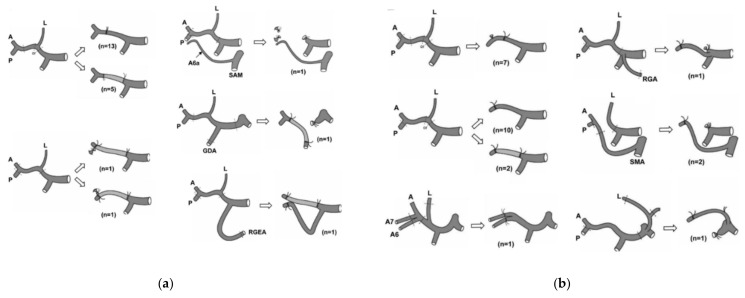
(**a**): Scheme of vascular resection for left hepatectomy. Type of hepatic artery resection and reconstruction during left hepatectomy. L: left hepatic artery; A: right anterior hepatic artery; P: right posterior hepatic artery; SMA: superior mesenteric artery; GDA: gastroduodenal artery; RGEA: right gastroepiploic artery; and A6a: artery supplying a small area of subsegment 6. (**b**) Hepatic artery resection and reconstruction in case of left trisectionectomy. L: left hepatic artery; A: right anterior hepatic artery; P: right posterior hepatic artery; RGA: right gastric artery; MA: superior mesenteric artery; A6: artery feeding segment 6; and A7: artery feeding segment 7. Note that two patients underwent arterioportal shunting because arterial reconstruction was technically impossible (Reproduced with permission from Ref. [15]. 2010, LWW).

**Table 1 cancers-14-02672-t001:** Cases of concomitant hepatic artery resection for perihilar cholangiocarcinoma reported in the English literature before 2010.

Author	Year	Number of Patients	Morbidity/Mortality	Long Term Prognosis
Tsuzuki [24]	1983	HA + PVR:2	N.S/0%	15 M and 18 M
Lee [25]	2000	HA: 4	N.S/N.S	N.S
Yamanaka [26]	2001	11	N.S/18%	N.S
Shimada [27]	2003	(PHCC/GBC) HA + PVR:6 HA:6	N.S/16.7% N.S/0%	N.S
Sakamoto [29]	2006	6	N.S/0%	4 M to 31 M
Miyazaki [28]	2007	9	78%/33%	MST:213 days

PHCC: perihilar cholangiocarcinoma; GBC: gallbladder carcinoma; M: month; MST: median survival time; N.S: not shown; HA: concomitant hepatic artery resection; and PVR: portal vein resection.

**Table 2 cancers-14-02672-t002:** Cases of concomitant hepatic artery resection for perihilar cholangiocarcinoma reported after 2010.

Author	Year	Number of Patients	Morbidity/Mortality	Long Term Prognosis
Nagino [15]	2010	50 (APS:2)	54%/2%	5 years: 30.3%
Wang [16]	2015	24	41.7/4.2%	5 years: 25%
Noji [5]	2016	28 (APS:11)	51.3%/3.6%	5 years: 25.5%
Matsuyama [17]	2016	44	66%/9%	5 years: 22.3%
Peng [18]	2016	26	16.1%/8.6%	5 years: 38.5%
Hu [19]	2018	29 (reconstruction) 34 (non-reconstruction)	20.7%/3.4% 17.6%/2.9%	5 years: 19% 5 years: 25%
Schimizzi [12]	2018	12	66%/N.S	MST: 33 months
Higuchi [20]	2019	19	47%/16%	5 years: 16%
Mizuno [3]	2020	146	49%/2.1%	5 years: 24.6%
Shindo [21]	2021	13	69.2%/15.4%	5 years: 0% (MST: 20.9 month)
Kuriyama [30]	2021	17	47.1%/11.8%	5 years: 26.9%
Sugiura [31]	2021	HA + PVR:17 HA:31	42%/6% 52%/0%	N.S

APS: arterial porta shunting; MST: median survival time; N.S: not shown; HA: concomitant hepatic artery resection; and PVR: portal vein resection.

## Data Availability

The data presented in this study are available in the present article and Appendix A.

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
