# Peer review of "Concomitant Hepatic Artery Resection for Advanced Perihilar Cholangiocarcinoma: A Narrative Review"

_cancers, 2022, doi:10.3390/cancers14112672_

Round 1

Reviewer 1 Report

This interesting review entitled "Concomitant hepatic artery resection for advanced perihilar cholangiocarcinoma: a narrative review" is a narrative review on a rare tumor such as advanced Kaltsin's tumor. Curative surgical resection is always the treatment of choice although the role of surgery is the focus of debate for locally advanced tumors with vascular/arterial infiltration.  Arterial infiltration is a common condition because of the anatomic location of the right hepatic artery, which in 75% of cases is located immediately behind the bile duct. Recent developments in operative procedures and perioperative management may have made resection at T4 (with involvement of the contralateral hepatic artery) possible.

However, the work definitely needs improvement in some aspects. 
The tables are poor, especially the first one. I would suggest to modify the first table or to add a different one. 
I find that the bibliography needs to be expanded because it is too poor.  I cannot find anything about the role of liver transplantation in locally advanced tumors, I suggest to quote a recent review published on a prestigious journal: doi: 10.3390/cancers13153657.  Equally interesting would be to cite the comparative results between liver transplantation and resection to have a more complete view of the subject: doi: 10.1097/SLA.000000002624. 

With appropriate modifications I find this article to be of great interest to the scientific community. 
I commend the authors for the straightforwardness of the discussion. 

Author Response

Thank you for reviewing our review paper. We have read your comments carefully and revised the manuscript accordingly. Before resubmission, the paper has also been reviewed once more by an academic English editing service.

Reviewer 1

This interesting review entitled "Concomitant hepatic artery resection for advanced perihilar cholangiocarcinoma: a narrative review" is a narrative review on a rare tumor such as advanced Kaltsin's tumor. Curative surgical resection is always the treatment of choice although the role of surgery is the focus of debate for locally advanced tumors with vascular/arterial infiltration.  Arterial infiltration is a common condition because of the anatomic location of the right hepatic artery, which in 75% of cases is located immediately behind the bile duct. Recent developments in operative procedures and perioperative management may have made resection at T4 (with involvement of the contralateral hepatic artery) possible.

However, the work definitely needs improvement in some aspects.

The tables are poor, especially the first one. I would suggest to modify the first table or to add a different one.

Author reply

Thank you for your comment. We believe that it was important to show the history of hepatic artery resection for perihilar cholangiocarcinoma. Accordingly, we believe that the data shown in Table 1 is indispensable to this review paper. Nevertheless, we have made several corrections to Table 1.

I find that the bibliography needs to be expanded because it is too poor. 

Author reply

We agree with your comment. We have added extra text discussing liver transplantation and ALPPS procedure for perihilar cholangiocarcinoma, as well as future perspectives on the field. The editorial office had also suggested that we expand the volume of the manuscript.

I cannot find anything about the role of liver transplantation in locally advanced tumors, I suggest to quote a recent review published on a prestigious journal: doi: 10.3390/cancers13153657.  Equally interesting would be to cite the comparative results between liver transplantation and resection to have a more complete view of the subject: doi: 10.1097/SLA.000000002624.

Author reply

Thank you for your suggestions. We have read the excellent review paper you recommended and cited it accordingly.

With appropriate modifications I find this article to be of great interest to the scientific community.

I commend the authors for the straightforwardness of the discussion.

Reviewer 2 Report

This narrative review analyses the role of arterial resection in case of T4 PHCC with contralateral hepatic artery involvement. This is an interesting topic and the debate on this technical possibility has recently gained attention because few pioneering centres reported oncological outcomes similar to those of resected patients without arterial encasement. Therefore, since the outcomes of non-surgical management of PHCC are poor and radical resection remains the optimal therapeutic option, technical advancements in surgical techniques are important in order to increase the pool of patients amenable to radical surgical resection.  The authors review the current available reports on arterial resection for PHCC and revise the technical options for arterial reconstruction (including a demonstrative video).

The manuscript is of interest and it is a useful summary of the current evidences on a technical possibility which probably is about to be investigated soon in high specialized centres.

There are few English language and minor spell check required but the manuscript is well written.

Minor comment:

  • In Fig 1 Numerals for the right segments are reported as 58 for the right anterior sector and 67 for the right posterior sector please modify with 5-8 and 6-7 respectively. In addition, the authors probably referred to Couinoud’s hepatic segments and not Cournand’s hepatic segments.
  • Conclusion: Please correct the final sentence in order to make it clearer to the readers. For example: … “Recent developments in operative procedures and perioperative management might have made resection of T4 PHCC with contralateral hepatic artery involvement a suitable option in order to pursue a radical resection. Hepatic resection associated to contralateral HAR for locally advanced PHCC is feasible and would be pursued in selected patients operated in high volume centres.”  

Author Response

Thank you for reviewing our review paper. We have read your comments carefully and revised the manuscript accordingly. The paper has also been reviewed once more by an academic English editing service before resubmission.

This narrative review analyses the role of arterial resection in case of T4 PHCC with contralateral hepatic artery involvement. This is an interesting topic and the debate on this technical possibility has recently gained attention because few pioneering centres reported oncological outcomes similar to those of resected patients without arterial encasement. Therefore, since the outcomes of non-surgical management of PHCC are poor and radical resection remains the optimal therapeutic option, technical advancements in surgical techniques are important in order to increase the pool of patients amenable to radical surgical resection.  The authors review the current available reports on arterial resection for PHCC and revise the technical options for arterial reconstruction (including a demonstrative video).

The manuscript is of interest and it is a useful summary of the current evidences on a technical possibility which probably is about to be investigated soon in high specialized centres.

There are few English language and minor spell check required but the manuscript is well written.

Minor comment:

  • In Fig 1 Numerals for the right segments are reported as 58 for the right anterior sector and 67 for the right posterior sector please modify with 5-8 and 6-7 respectively. In addition, the authors probably referred to Couinoud’s hepatic segments and not Cournand’s hepatic segments.

Author reply

Thank you for your comment. We have made the required corrections to Figure 1.

  • Conclusion: Please correct the final sentence in order to make it clearer to the readers. For example: … “Recent developments in operative procedures and perioperative management might have made resection of T4 PHCC with contralateral hepatic artery involvement a suitable option in order to pursue a radical resection. Hepatic resection associated to contralateral HAR for locally advanced PHCC is feasible and would be pursued in selected patients operated in high volume centres.”  

Author reply

Thank you for your correction. We have revised the conclusion in accordance with your suggestion.

Round 2

Reviewer 1 Report

Good revision: the paper is suitable for publication

Best regards